# A Sparse and Low-Rank Regression Model for Identifying the Relationships Between DNA Methylation and Gene Expression Levels in Gastric Cancer and the Prediction of Prognosis

**DOI:** 10.3390/genes12060854

**Published:** 2021-06-02

**Authors:** Yishu Wang, Lingyun Xu, Dongmei Ai

**Affiliations:** 1School of Mathematics and Physics, University of Science & Technology Beijing, Beijing 100083, China; 2School of Mathematics and Statistics, Qingdao University, Qingdao 266003, China; x15606482895@163.com; 3Basic Experimental Center of Natural Science, University of Science and Technology Beijing, Beijing 100083, China

**Keywords:** low-rank sparse regression model, DNA methylation, prognosis, gene expression, gastric cancer

## Abstract

DNA methylation is an important regulator of gene expression that can influence tumor heterogeneity and shows weak and varying expression levels among different genes. Gastric cancer (GC) is a highly heterogeneous cancer of the digestive system with a high mortality rate worldwide. The heterogeneous subtypes of GC lead to different prognoses. In this study, we explored the relationships between DNA methylation and gene expression levels by introducing a sparse low-rank regression model based on a GC dataset with 375 tumor samples and 32 normal samples from The Cancer Genome Atlas database. Differences in the DNA methylation levels and sites were found to be associated with differences in the expressed genes related to GC development. Overall, 29 methylation-driven genes were found to be related to the GC subtypes, and in the prognostic model, we explored five prognoses related to the methylation sites. Finally, based on a low-rank matrix, seven subgroups were identified with different methylation statuses. These specific classifications based on DNA methylation levels may help to account for heterogeneity and aid in personalized treatments.

## 1. Introduction

Gastric cancer (GC) is a cancer of the digestive system, and it has a high mortality rate globally, ranking second among all cancers [1]. In recent years, GC morbidity and mortality rates have increased because of changes in our diet and environment. Patients are often diagnosed with GC at an inoperable stage, and recurrence is common after resection [1]. Tumor heterogeneity allows for early diagnosis and effective treatment of patients with different types of GC. GC is highly heterogeneous and shows varying sensitivity to chemotherapy among different clinical subtypes. Therefore, molecular oncology studies of GC are urgently needed to obtain better prognostic outcomes. GC usually originates from Helicobacter pylori infection, which may lead to chronic inflammation and consequent tumorigenesis [2]. In addition, other risk factors have been identified, such as environmental, genetic, and epigenetic factors [3]. Because of tumor heterogeneity, the same treatment can result in different prognoses for different GC subtypes. Identification of the genetic heterogeneity of GC could help clinicians to better understand the mechanism of GC. 

The functional heterogeneity of cancer cells within tumors involves weak and varying genetic expression between cells or different functional cell subpopulations [4], indicating a more comprehensive understanding of tumor cells. One basic aspect of cancer cell heterogeneity in the same tumor is the different levels of gene expression, which may be influenced by many factors, for example, epigenetic changes which include DNA methylation, non-coding RNA, chromatin remodeling, and histone modifications [3]. Epigenetic processes can regulate the expression of genes but without DNA sequencing changes and is inheritable across generations. DNA methylation is a well-characterized epigenetic modification that plays an important role in carcinogenesis [5,6], and is mediated by DNA methyltransferase. DNA hypermethylation at promoter regions can silence the expression of targeted genes [7], further influencing cell cycles, DNA repair, and even the signaling pathways of tumor development. It plays an important role in promoting cancer [6,8,9]. However, not all CpG sites regulate gene expression. Only hypermethylation of CpG islands in specific regions inhibits the expression of tumor suppressor genes (TSGs), DNA repair genes, housekeeping genes, and cell cycle control genes. Currently, exploring the relationships between gene expression levels and DNA methylation regions based on a specific tumor is needed for tumor typing and prognosis prediction. 

For GC, although several CpG sites are involved in the processes of GC development [3], there is a lack of systematic analysis to establish an efficient model that can provide insights into low gene expression levels in different GC subtypes affected by specific DNA methylation sites, as well as further help to prioritize disease-associated methylation sites and contribute to GC heterogeneity. Most abnormal changes, including methylation or demethylation of specific DNA methylation sites, exist in different GC TSGs or oncogenes. These changes often occur before tumor formation or development. Accordingly, if their sites can be accurately confirmed, they can be considered to be early diagnostic GC markers or predictors for people with a high risk of GC. 

In contrast, high concordance in methylation and gene expression predicts tumor immune infiltration levels [10] and immune-informative CpG sites show significant prognostic value [10]. Therefore, identification of specific CpG sites associated with disease would allow the prediction of relationships between methylation CpG sites and cancer prognosis, which would provide a reference for clinical practices of cancer prognosis. One approach is to determine which aspects of DNA, such as gene expression levels, are affected by DNA methylation. Naively, such association can be identified using a simple statistical test on all paired combinations of DNA methylation and gene transcripts.

However, gene expression levels are influenced by many factors, including DNA methylation, changes in DNA sequence, and other hidden factors, such as cellular state [11], environmental factors [12], and experimental conditions [13]. The interaction of most epigenetic variations with genetic variation is diverse. Moreover, a wide variety of confounders lie hidden in the data, leading to both spurious associations and missed associations if not properly addressed. An interesting problem is how to distribute the linkage between epigenetic and genetic variations. In particular, an association analysis of these RNA-directed DNA methylation regions with genetic variants could identify the methylation quantitative trait loci, which are related to GC.

In this study, we introduce an alternative formulation to address these problems. We make use of sparse regression for RNA-directed DNA methylation mapping and propose a low-rank representation to account for the DNA sequence changes and other nongenetic hidden factors. Our methodology is inspired by the following linear mixed model [14]: y=Xβ+Zμ+e, where y is the vector of data; β is a vector of fixed effects; μ is a vector of random effects; and X and Z are the model matrices corresponding to β and μ, respectively. This linear mixed model has been used to model the hidden factors in eQTL mapping problems [15].

We aimed to establish the relationship between DNA methylation and genetic variation (here, we focused on gene expression levels) related to GC subtypes. We rewrote this mixed model because it could be separated passively. We used sparse regression $X_{1}\beta_{1}$ to account for DNA methylation mapping, which was related to differential gene expression levels. We put the DNA sequence changes and hidden factors related to GC tumor subtypes, separately, into $X_{2}\beta_{2}$ and Zμ. Then, we merged these two parts into one low-rank matrix representation, L, for the expression heterogeneity (EH). Thereby, the regression model could be rewritten as follows: y=Xβ+L+e. Because a low-rank matrix can always be written as L=WH, our model can be regarded as an equivalent form of the linear mixed model. 

There are two main advantages of our model as follows: Multiple gene expression levels and confounder effects can be jointly analyzed in disease-associated DNA methylation site studies, and the low-rank matrix, **L**, represents different patients’ sample clusters according to the confounder effects. Then, we can identify the differentially expressed genes of the GC subtypes, which also have specific methylation sites.

## 2. Materials and Methods

### 2.1. Methods

Let **Y** be a n×q matrix corresponding to a gene expression dataset where n is the number of samples and q is the number of genes. Let X be an n×q matrix corresponding to a DNA methylation dataset, where p is the number of DNA methylation sites. To model the relationship between **Y** and **X**, we propose decomposing **Y** as follows:(1)Y=XB+L+e
where $B\in R^{p\times q}$ is the coefficient matrix and $e\in R^{n\times q}$ is a Gaussian random noise term with zero mean and variance $\sigma^{2},\;i.e.,\;e_{ij}\sim N(0,\sigma^{2})$. Here, we introduce $L\in R^{n\times q}$ to our model to account for the variations caused by EH, including DNA sequence changes and other hidden nongenetic factors. This model implies that gene expression levels are influenced by DNA methylation, EH is related to tumor subtypes, and random noise.

To make the decomposition (1) possible, we made the following assumptions:There are only a few expression heterogeneous factors related to tumor subtypes that influence differential gene expression levels. Thus, **L** is a low-rank matrix. Then, we can obtain gene clusters according to the matrix **L**.The gene expression level is only affected by a small fraction of DNA methylation sites. This implies that the coefficient matrix **B** should be sparse.

On the basis of these assumptions, Formula (1) can be rewritten as per the following optimization problem:(2)$$\begin{array}{l} \underset{B,L}{\min}\|Y-XB-L{\|}_{F}^{2}\\ s.t.\;rank(L)\leq r_{0},\;\|B{\|}_{1}\leq t_{0} \end{array}$$
where $|B{\|}_{1}$ is the element-wise $l_{1}$ norm and $|W{\|}_{F}^{2}=\sqrt{\sum_{ij}W_{ij}{}^{2}}$ is the Frobenius norm. To make this minimization problem a convex surrogate, we relax the rank operator on **L** with the nuclear norm, which has been previously proven to be effective [16], as follows:(3)$$\underset{B,L}{\min}\frac{1}{2}\|Y-XB-L|{|}_{F}^{2}+\rho \|B{\|}_{1}+\lambda \|L{\|}_{*}$$
where $\|L{\|}_{*}$ is the nuclear norm of **L**. *ρ* and λ are regularization parameters that control the sparsity of **B** and the rank of **L**.

Then, we adopted an alternating strategy to solve problem (3).
For fixed **B**, the optimization problem becomes:
(4)$$\underset{L}{\min}\frac{1}{2}\|Y-XB-L|{|}_{F}^{2}+\lambda \|L|{|}_{*}$$

The solution of **L** is Formula (5), according to [16], as follows:(5)$$L=S_{\lambda }(Y-XB)$$
where $S_{\lambda }(W)=UD_{\lambda }V^{T},\;with\;D_{\lambda }=diag[{\left(d_{1}-\lambda \right)}_{+},\ldots ,{\left(d_{r}-\lambda \right)}_{+}]$, and $UDV^{T}$ is the singular value decomposition (SVD) of W, $D=diag[d_{1},\ldots ,d_{r}]$, and $t_{+}=\max(t,0)$.

For fixed L, the optimization problem becomes:(6)$$\underset{B}{\min}\frac{1}{2}\|Y-XB-L|{|}_{F}^{2}+\rho \|B|{|}_{1}$$

It can be decomposed into *q* independent Lasso problems [17] as follows: $$\underset{B_{j}}{\min}\frac{1}{2}\|Y_{j}-XB_{j}-L_{j}{\|}_{F}^{2}+\rho \|B_{j}{\|}_{1},\;j=1,\ldots ,q$$
where $Y_{j}$, $L_{j}$, and $B_{j}$ are the *j*th columns of **Y, L**, and **B**. The Lasso problem can be solved efficiently by the coordinate descent algorithm [15,18].

After the two alternative steps, we selected the top *d* DNA methylation sites for each gene (based on the absolute value of the coefficients) and the low-rank matrix **L**. Then, we built the methylation prognosis subtypes based on specific methylation sites, and relationships between the methylation sites and gene expression levels. Meanwhile, the low-rank matrix **L** also provided a classification of tumor samples that may indicate the different GC subtypes caused by different methylation sites and epigenetic levels, which was defined as EH in our model. 

### 2.2. Synthetic Data

To demonstrate the effectiveness of our model, while avoiding the simulation setup favoring it, we generated the synthetic data, as in the setup in [19], as follows: For the methylation effects, each methylation site is generated independently and uniformly from a binomial distribution with the probability *p* = 0.25 denoted by matrix **X** with dimension n×p. The coefficient matrix **B** is a sparse matrix with dimension p×q, with 2% non-zero entries, which are generated using a standard Gaussian distribution. Let G denote the methylation effect **G** = **XB**.**EH**: The covariance matrix is Σ generated by $HH^{T}$, with $H\in R^{n\times K}$ and $H_{i,j}\sim N(0,1)$. Here, K is the number of hidden factors. The random variable $L_{j}$ was drawn from N(0,τΣ). Let $L=[L_{1},\ldots ,L_{q}]$Let $e\sim N(0,\sigma_{e}^{2}I)$

Now the synthetic data are: (7)Y=XB+μ+e=G+μ+e

### 2.3. Gastric Cancer Data

The GC datasets were downloaded, including the mRNA expression, methylation, and clinical datasets, from The Cancer Genome Atlas repository (http://portal.gdc.cancer.gov/, aceessed on 2 June 2021).

## 3. Results

### 3.1. Synthetic Results Demonstrated Our Model Benefited High Dimensional Data

We used the synthetic dataset, constructed as described in the Materials and Methods, to test the performance of our method under different settings, i.e., we varied *n, p,* and *q*. Figure 1 shows the receiver operating characteristic curves of different settings. From the simulation results, we observed that our method is beneficial with high dimensional data; is robust with a larger *q*, which is the dimension of the response variable in the regression model; that the higher the number of samples, the better the regression results; and if the number *p* is larger than *q* and *n*, this may result in overfitting.

### 3.2. Gastric Cancer Transcription and Methylation Datasets Filtration

We downloaded the gastric cancer dataset from The Cancer Genome Atlas repository (http://portal.gdc.cancer.gov/, aceessed on 2 June 2021). There were 407 patients with 375 tumor samples and 32 normal samples with 56,753 gene expression profiles for each sample in this gastric cancer dataset and 469 patients with 443 tumor samples and 27 normal patients with 19,755 DNA CpG sites. First, in more than 70% of the samples, we deleted the methylation sites that had missing values; these were located in the second chromosome, and CpGs from above 2 kb upstream to 0.5 kb downstream (gene promoter regions). The clinical samples were also filtered if the follow-up duration was less than 30 days, or there were no follow-up data. Then, we took the intersection of samples based on the mRNA expression and methylation datasets to use in our model. There were 378 patients with 351 tumor samples and 27 normal patients. Furthermore, the number of gene expression profiles was too large to provide significant information. We selected the differentially expressed genes between tumor and normal samples as response variables.

### 3.3. Application of Our Model to Gastric Cancer Dataset

Here, the Y matrix was read from the mRNA expression dataset, and X was read from the DNA methylation dataset. Our regression-decomposed model gave two matrixes, i.e., B and L, where B denoted the relationships between DNA methylation sites and gene expression levels and L denoted a low-rank matrix. Similar genes shared the same rank. Figure 2 shows the linkage peaks in this GC study given by our model; the top 1000 associations based on abs (B) are shown here. Figure 3A shows the methylation levels of the top 100 methylation sites based on abs (B) and Figure 3B shows gene expression levels of the genes corresponding to the 100 sites. These 100 CpG sites indicated the important influence of DNA methylation on gene expression in GC samples. In addition, Figure 4 shows the gene ontology (GO) (A) and the Kyoto Encyclopedia of Genes and Genomes (KEGG) (B) enriched functional items of these methylation sites (*p*-value of <0.05). Most of these sites were associated with carcinogenesis, metabolism, and ECM-receptor interaction functions and most of the genes corresponding to them took part in receptor ligand activity and collagen-containing extracellular pathways.

### 3.4. 29 Genes Were Identified as Gastric Cancer-Associated Methylation-Driven Genes

To find more information related to DNA methylation sites with gene expression levels, we used R package “Methylmix” [20] to find the methylation-driven genes with low mRNA expression and high methylation levels and negative correlations between methylation and expression in cancer samples (*p* < 0.05). There were 31 genes selected by “Methylmix,” and 29 genes overlapped with those corresponding to 100 associations based on abs (B). These genes were RPP25, PLXNC1, BOP1, HOXC13, BST2, TMEM26, ARHGAP20, TONSL, CLEC7A, STC2, HOXA13, MCMDC2, DNAH14, PRELID1P1, HOXA11, "DPY19L1, CDKN2A, GPR84, ZNF525, FAM24B, ZFPM2-AS1, RANBP17, C3AR1, SAC3D1, RPS6KA6, PIWIL1, CCNI2, BZW2, and PALB2. They were the important methylation-driven genes associated with GC whose corresponding methylation sites might affect gene expression levels. The other two genes HOXA10-AS and HOXA11-AS, which were not recognized by our model had been demonstrated do not have obvious differential methylation levels between tumor and normal samples in Figure 5. Figure 6 shows the GO and KEGG functional items of these genes with different log FC values, which were denoted as upregulated if log FC > 0, otherwise, they were denoted as downregulated. Here, we defined upregulated genes as having higher methylation levels in cancer subtypes and lower methylation levels in normal subtypes, otherwise, they were downregulated genes. Figure 5 shows the methylation and gene expression level heatmap results of the 31 methylation-driven genes, where the genes HOXA10-AS and HOXA11-AS do not have obvious differential methylation levels between tumor and normal samples. In the Appendix A, we also provide the correlation relationship results between the 29 GC-associated methylation-driven gene expression levels and methylation levels Appendix A.

### 3.5. Methylation Markers Associated with Prognosis of GC

For patients with GC, one important problem is poor prognosis because of different GC subtypes, with complex genetic heterogeneity. In this study, we aimed to identify specific biomarkers that could help clinicians to better understand the mechanism of GC. We merged the clinical data with the mRNA expression profiles and DNA methylation profiles of these 29 methylation-driven genes associated with GC. A prognostic assessment model was established, and the Cox risk regression model was used to identify independent prognostic methylation sites, implemented by using the R package “survival” [21]. First, using univariate Cox regression analysis, we obtained 17 methylation sites related to patient prognosis (*p* < 0.05). Then, a multivariate Cox regression was used to analyze these sites. Five methylation CpG sites were observed, and the risk score was computed. The genes that corresponded to these CpG sites were cg01531665 (NOL6), cg01201519 (FEZF1), cg00484488 (ADPGK), cg00730266 (PPP1R14A), and cg00333849 (KLHL35), which were prognosis-related sites. Figure 7A shows the methylation heatmap of tumor samples based on these five genes and Figure 7B shows the heatmap of risk scores based on these five genes for all tumor samples and the grouping results of samples with low or high risk. Next, according to the risk score, we grouped the patients into high- and low-risk groups if their risk scores were higher than the median of all risk scores. The risk assessment showed that high-risk patients had a poorer prognosis than low-risk patients. The methylation levels of the selected sites in the established model decreased as the risk scores increased. Figure 7C shows the survival curve with different risk subgroups. Figure 7D shows the hazard ratio of the risk score to other multi-covariates, such as age, gender, grade, and cancer stage. According to this hazard ratio, the risk score based on these methylation CpG sites could be regarded as a prognostic factor.

### 3.6. Exploration of the Subtype-Associated GpG Sites 

On the one hand, the GC molecular subtype seemed to influence the prognoses of the patients in this study, while, on the other hand, the expression heterogeneity (EH) in our model indicated the different GC subtypes caused by different methylation sites and epigenetic levels. In order to explore the subtype-associated CpG sites based on classifications of tumor samples, we used the low-rank matrix **L** as classification features to obtain the samples’ classifications. Because **L** is low rank, variables sharing the same rank could be regarded as being in one group [22]. In our algorithm, when optimal problems converge, the rank of matrix **L** is seven. Therefore, we grouped GC samples into seven clusters according to their ranks. A heatmap of the methylation levels of GC subtypes is shown in Figure 8A. The boxplot of methylation levels based on these seven clusters is shown in Figure 8B. According to the clustering results obtained by matrix **L**, we evaluated all methylation sites for different clusters, finding eight subtype-associated CpG sites which had differential methylation levels in more than two clusters with *p*-values <e-10 (cg01201519 (FEZF1), cg00388897 (KLHL35), cg01531665 (NOL6), cg02422011 (PPP1R14A), cg00563678 (RNF150), cg01192487 (SAMD12), cg00328900 (VCAN), and cg00112309 (ZFPM2)).

### 3.7. Association between the Eight Subtype-Associated CpG Sites and Prognosis

We constructed the prognosis survival curves for all the GC samples grouped by hypermethylation with low expression and hypomethylation with high expression of the eight subtype-associated CpG sites (Figure 9). In this figure, the Kaplan–Meier curves show that the prognostic results differed significantly between the two groups. 

### 3.8. Identifying Four Subtype-Specificity Prognosis-Related Sites 

Finally, the intersection analysis between the five prognosis-related sites and the eight subtype-associated sites was carried out, and four factors, i.e., cg01531665 (NOL6), cg01201519 (FEZF1), cg02422011 (PPP1R14A), and cg00388897 (KLHL35) were found to be overlapped. These four CpG sites were prognosis-related sites that influenced gene expression levels and also were related to GC subtypes, denoted as subtype-specificity prognosis-related sites. Gene ontology (GO) and the KEGG items of genes according to these four CpG sites are shown in Table 1. Then, we used an online analysis tool “MEXPRESS” (https://mexpress.be/, aceessed on 2 June 2021) to verify the correlations among the expression levels of these four genes and DNA methylations. Figure 10A–D show the results of different genes. On the right of each subfigure, the r values are the Pearson correlation coefficients between one specific gene expression level and DNA methylation levels of different methylation sites. Meanwhile, each subfigure also shows the relationships of other factors such as sample type, tumor stage, and gender.

### 3.9. Determining the Influential Power of the Subtype-Specificity Prognosis-Related Sites on Expression Levels—Important Prognostic Markers and Regulation of Gene Expression Factors

It can be observed that there is one position of the gene NOL6 that had a significantly positive correlation with its expression (Figure 10A), 26 positions of the gene FEZF1 CpG site have significantly negative correlations with its expression, while 12 CpG sites have significantly positive correlations with its expression. Actually, it has been demonstrated that the gene FEZF1-AS1 (FEZF1 antisense RNA) can epigenetically repress the expression of P21 which is a demethylase [23]. Studies have found that the gene FEZF1-AS1 can act as an “oncogene” for gastric cancer partly through suppressing P21 expression and may serve as a candidate prognostic biomarker for new therapies of gastric cancer patients. Simultaneously, there were 14 positions of the gene PPP1R14A CpG site that had significantly negative correlations with its expression, while there were 12 positions of the gene KLHL35 CpG site that had significantly negative correlations with its expression. In addition, the gene PPP1R14A, which is regulated by promoter region methylation, has been proven to play a key role in the initiation and progression of gastric cancer, colorectal cancer, and lymphomas [24,25,26]. At the same time, hypermethylation of the gene KLHL35 has also been demonstrated to be associated with the development of hepatocellular carcinoma [27,28]. These results demonstrate the predictive accuracy of our model, which can identify the significant CpG sites that have an influence on gene expression, meanwhile, exploring prognosis-related sites associated with GC subtypes. These results demonstrated that the four CpG sites were not only associated with the GC subtypes and patients prognosis, but also had significant correlation with the gene expression levels. It means that they had potential to be important prognostic markers and regulate of gene expression factors.

## 4. Discussion

Epigenetic changes, such as DNA methylation, are closely associated with the development and malignant transformation of GC. DNA methylation changes accumulated as oncogenesis may result in different GC subtypes. Therefore, identification of specific DNA methylation sites which associated with gene expression and prognosis of tumor patients, might aid in the development of personalized treatment plans. In this study, we introduced a linear regression model to prioritize the relationship of disease-associated methylation sites and gene expression levels in different sample groups, and then provided an epigenetic explanation of tumor heterogeneity. 

Our model made use of matrix decomposition theory by introducing a sparse regression to account for DNA methylation mapping and a low-rank matrix to contain the expression heterogeneity (EH), resulting in one sparse coefficient matrix **B** and one low-rank matrix **L**. According to the top abs (B), we obtained the specific DNA methylation sites that were related to GC related to different gene expression levels. Because our model does not perform statistical significance tests, we were not able to report our results based on statistical significance. Alternatively, we were more interested in the top signals. Thus, we only showed the top 1000 associations based on the absolute value of **B** in Figure 2 and selected the top 100 methylation sites for subsequent research. Concurrently, we used the low-rank matrix **L** to obtain seven tumor sample clusters, from which we selected eight subtype-associated CpG sites.

In this article, our study firstly identified 29 Gastric Cancer-associated methylation-driven genes, whose corresponding methylation sites might affect gene expression levels, based on the top 100 abs (B). Then we used the Cox risk regression model to obtain five important prognosis-associated methylation sites that can be regarded as prognostic factors: cg01531665 (NOL6), cg01201519 (FEZF1), cg00484488 (ADPGK), cg00730266 (PPP1R14A), and cg00333849 (KLHL35). By using them to compute the risk score to predict the survival curves, we classified patients into high-risk and low-risk groups, in which patients with high-risk scores were corresponding to poor prognosis, and on thecontrary, low risk corresponded to relatively good prognosis. 

Then according to the low-rank matrix **L**, our model explored seven Gastric Cancer subtypes, by which we can evaluated all methylation CpG sites with the methylation levels to find the differentially methylated sites among different GC tumor groups. In this way, we identified eight subtype-associated CpG sites: cg01201519 (FEZF1), cg00388897 (KLHL35), cg01531665 (NOL6), cg02422011 (PPP1R14A), cg00563678 (RNF150), cg01192487 (SAMD12), cg00328900 (VCAN), and cg00112309 (ZFPM2). Kaplan-Meier curves demonstrated that these sites also had relationships with prognosis. Therefore, an intersection analysis between the five prognosis-associated methylation sites and the eight subtype-associated sites was conducted, resulting in four sites: cg01531665 (NOL6), cg01201519 (FEZF1), cg02422011 (PPP1R14A), and cg00388897 (KLHL35). The downstream analysis demonstrated that they were not only associated with the GC subtypes and patients prognosis, but also had a significant correlation with the gene expression levels, which indicated that they had potential to be important prognostic markers and regulate of gene expression factors.

Despite the advantages of our approach, the limitation is that we did not provide a rigorous statistical significance test of the estimated coefficient matrix **B**. Researchers can rank associations and select those that they are interested in.

## Figures and Tables

**Figure 1 genes-12-00854-f001:**
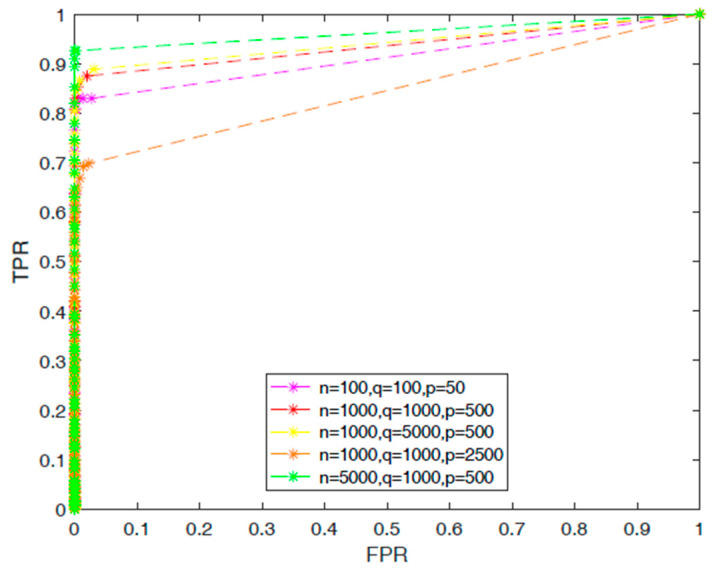
ROC curves of different parameters.

**Figure 2 genes-12-00854-f002:**
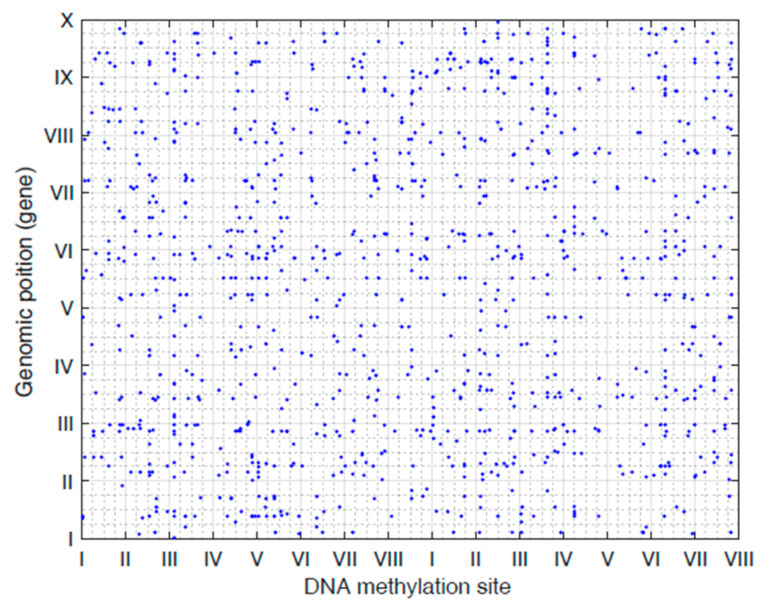
The plot of linkage peaks given by our model (Top 1000 associations based on abs(matrixB) are shown here).

**Figure 3 genes-12-00854-f003:**
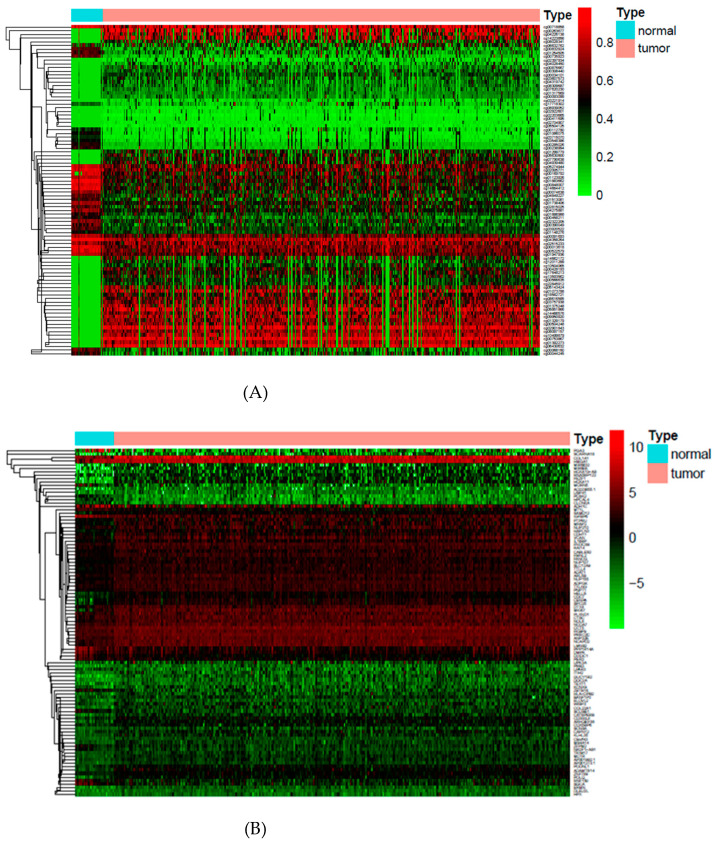
(**A**) The methylation levels of the top 100 methylation sites based on abs(B); (**B**) the gene expression levels of genes corresponding to these methylation sites.

**Figure 4 genes-12-00854-f004:**
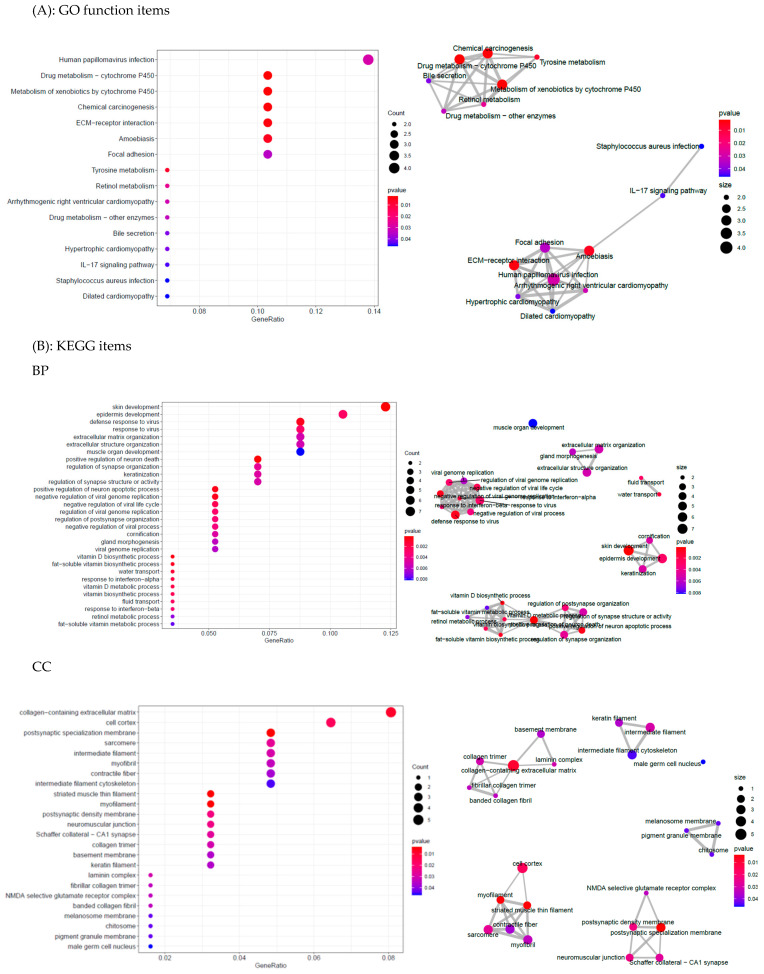
GO and KEGG enrichment of the top 100 methylation sites.

**Figure 5 genes-12-00854-f005:**
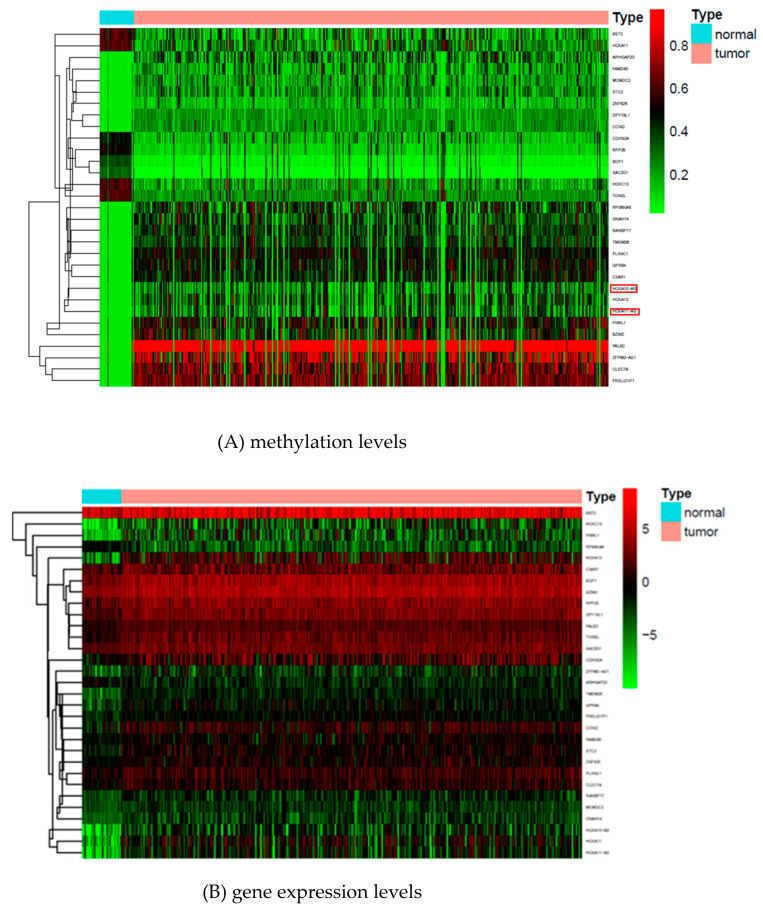
Heatmap of methylation and expression levels based on these 31 driven genes.

**Figure 6 genes-12-00854-f006:**
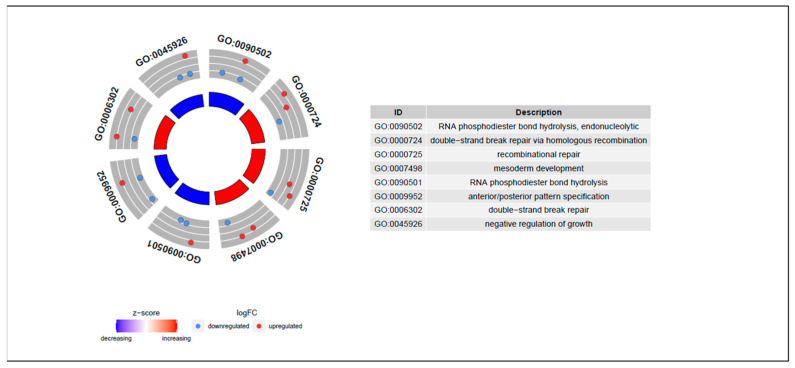
GO items of the driven genes.

**Figure 7 genes-12-00854-f007:**
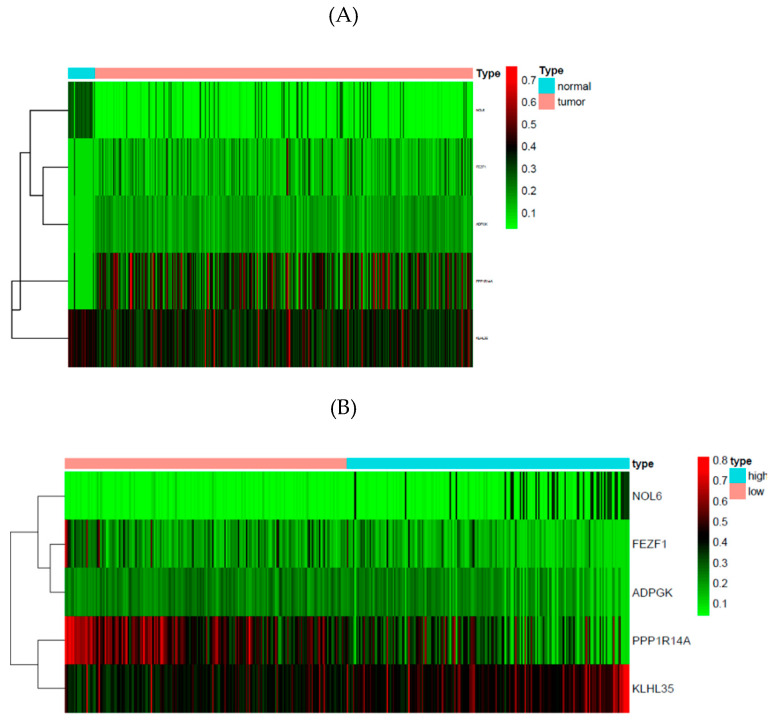
(**A**) Heatmap of methylation levels of samples based on five prognosis genes; (**B**) heatmap of risk Score based on five prognosis genes; (**C**) survival curves; (**D**) hazard ratio.

**Figure 8 genes-12-00854-f008:**
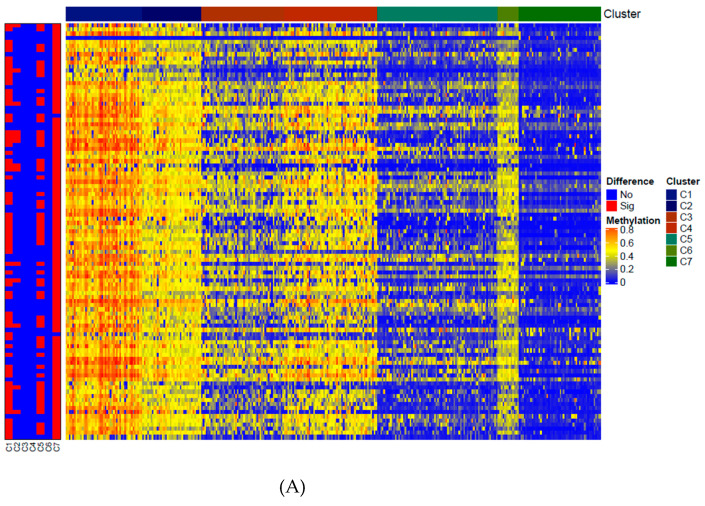
(**A**) Heatmap of methylation sites in different gastric cancer subtypes grouped by matrix L. The heat rectangle on the left map is red if there is a statistically significant difference between the selected cluster and the other clusters; otherwise, it is blue. Sig, significance, *p* < 0.05; No, no significance. (**B**) The box-plot of methylation levels of these seven clusters.

**Figure 9 genes-12-00854-f009:**
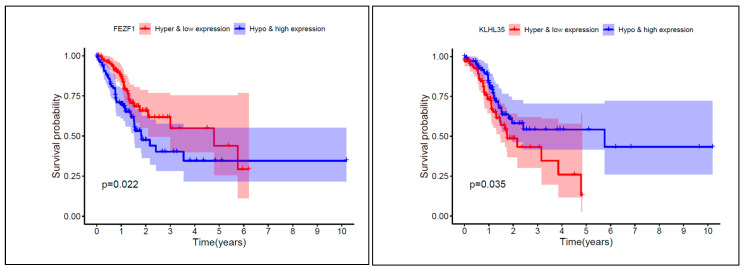
Prognosis survival curves of eight different expressed methylation sites.

**Figure 10 genes-12-00854-f010:**
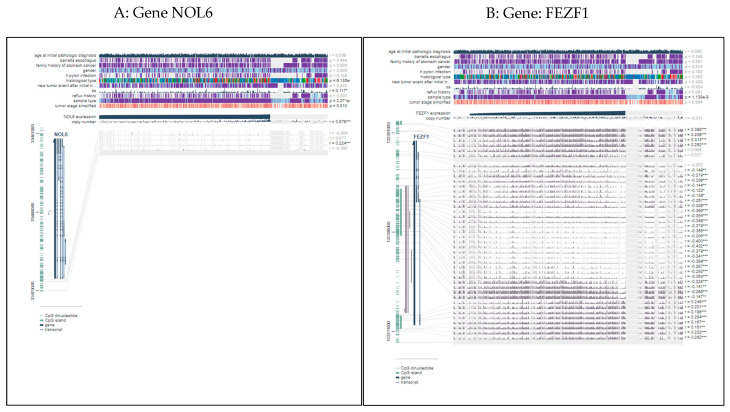
The results of correlation between various factors and gene expression, especially the relationship of gene methylation levels and expression level. (**A**) Gene NOL6 (**B**) Gene: FEZF1 (**C**) Gene PPP1R14A (**D**) Gene KLHL35

**Table 1 genes-12-00854-t001:** GO and KEGG items of NOL6, FEZF1, PPP1R14A and KLHL35.

Gene Name	GO	KEGG
FEZF1	negative regulation of transcription from RNA polymerase II promoter	None
KIHL35	protein binding	None
NOL6	rRNA processing, tRNA export from nucleus	Ribosome biogenesis in eukaryotes
PPP1R14A	regulation of protein dephosphorylation, cellular response to drug, phosphatase inhibitor activity	Vascular smooth muscle contraction

## Data Availability

All the data in our manuscript are available in The Cancer Genome Atlas repository (http://portal.gdc.cancer.gov/, aceessed on 2 June 2021). We also have uploaded them with the supplementary.

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
