# Peer review of "A Sparse and Low-Rank Regression Model for Identifying the Relationships Between DNA Methylation and Gene Expression Levels in Gastric Cancer and the Prediction of Prognosis"

_genes, 2021, doi:10.3390/genes12060854_

Round 1

Reviewer 1 Report

The authors answered my concerns and comments.

Author Response

Thank you for your suggestion

Reviewer 2 Report

  1. The authors say that "There were 406 patients with 375 tumor samples and 32 normal samples with 56,753 gene expression profiles for each sample in this gastric cancer dataset and 469 patients with 443 tumor samples and 27 normal patients with 19,755 DNA CpG sites." but why only "378 patients with 351 tumor samples and 27 normal patients" were chosen? What criteria were used?
  2. In extension to point (1), I encourage the authors to ask themselves - can anyone, anyone not involved in this current study, identify the samples from their description? I certainly cannot. Hence, how can their results be repeatable?

Round 2

Reviewer 2 Report

The authors addressed all of my concerns.